# Research

psychology

developmental prosopagnosia, Cambridge Face Memory Test, test–retest reliability, face recognition

**Author for correspondence:**
Ebony Murray
e-mail: emurray@bournemouth.ac.uk

# Diagnosing developmental prosopagnosia: repeat assessment using the Cambridge Face Memory Test

Ebony Murray and Sarah Bate

Department of Psychology, Bournemouth University, Poole, UK

EM, 0000-0003-4928-5871; SB, 0000-0001-5484-8195

Developmental prosopagnosia (DP) is a cognitive condition characterized by a relatively selective impairment in face recognition. Currently, people are screened for DP via a single attempt at objective face-processing tests, usually all presented on the same day. However, several variables probably influence performance on these tests irrespective of actual ability, and the influence of repeat administration is also unknown. Here, we assess, for the first known time, the test–retest reliability of the Cambridge Face Memory Test (CFMT)—the leading task used worldwide to diagnose DP. This value was found to fall just below psychometric standards, and single-case analyses revealed further inconsistencies in performance that were not driven by testing location (online or in-person), nor the time-lapse between attempts. Later administration of an alternative version of the CFMT (the CFMT-Aus) was also found to be valuable in confirming borderline cases. Finally, we found that performance on the first 48 trials of the CFMT was equally as sensitive as the full 72-item score, suggesting that the instrument may be shortened for testing efficiency. We consider the implications of these findings for existing diagnostic protocols, concluding that two independent tasks of unfamiliar face memory should be completed on separate days.

## 1. Introduction

Developmental prosopagnosia (DP) is a cognitive condition characterized by a relatively selective impairment in face recognition [1]. These difficulties occur in the absence of any neurological damage, socio-emotional dysfunction or lower-level visual deficits, and are estimated to affect 2–2.5% of the adult population [2] and 1.2–4% of those in middle childhood [3] (although note that, by definition, the lower end of a normal distribution would also encompass 2.5% of the population: [1,4]).

Over the past 20 years, research into DP has surged, and individuals with the condition have been used to make theoretical inferences about the development and functioning of the cognitive and neural architecture of the face recognition system (e.g. [5–9]). Given that these theoretical inferences rely on accurate identification of people with DP, and that some individuals also report moderate-to-severe psychosocial consequences of the condition [10–13], existing diagnostic protocols are under increasing scrutiny. Currently, most researchers diagnose the condition using a combination of the Cambridge Face Memory Test (CFMT: [14]), the Cambridge Face Perception Test (CFPT: [15]) and a famous faces test (e.g. [16–18]). Dominant recommendations suggest that DP is diagnosed when scores fall into the impaired range (typically at least two s.d. from the control mean) on any two of these three tasks [4,19].

This recommendation for repeated testing (i.e. not relying on a single score on a single test) reduces 'the chance that it happened by chance' [20, p. 945] and is vital for minimizing false alarms. However, this protocol does not allow for the possibility that the three assessment tasks measure different sub-components of face-processing that may be selectively impaired [21]. That is, while allowing for some overlap in processes on the three tasks (i.e. all three arguably tap face perception to some degree), it is broadly accepted that the CFMT assesses short-term, unfamiliar face memory [22], famous face tests measure long-term face memory and/or semantics and naming [23,24], whereas the CFPT is predominantly a measure of face perception [15]. By only diagnosing DP when two (or more) of these sub-processes are impaired, individuals who may be impaired on only one process are rejected (for example cases see [21]), and the possibility that intact performance may occur by chance on a single administration of each task is not accommodated.

Within-person performance instability can be observed for many reasons on everyday cognitive tasks [25], and empirical evidence suggests that short-term cognitive performance fluctuates considerably in children and younger adults, and even more so in older individuals [26–30]. Inconsistent performance has also been reported in the face recognition literature for typical participants and those with proficient face-processing skills [31–33]. Aside from the influence of chance [20,34] these inconsistencies could result from intrinsic factors (e.g. mood, fatigue or distraction) or the psychometric properties of the task in hand [35].

Fortunately, the face recognition literature has one 'gold standard' assessment task in the CFMT [2,36–38]. This test introduces six unfamiliar male faces for participants to memorize. They are then presented with triads of faces and are asked to segregate the target face from two distractors in (i) 18 trials as part of the encoding phase, (ii) 30 trials seen under novel lighting and viewpoint conditions, and (iii) 24 trials with added visual noise. The test substantially improves hit rates for diagnosing DP in comparison to other well-known face recognition tests [2,14,39], and shows a substantial inversion effect [15,40]. Correlation studies demonstrate the independence of CFMT performance from other cognitive abilities such as IQ [41,42], object recognition [16,43], verbal memory [38] and global attention [42]. However, the test–retest reliability of the CFMT has not yet been investigated, and it is currently unknown whether practice effects occur on repeat administration of the task.

'Test–retest reliability' refers to the investigation of consistency, reproducibility and agreement among two (or more) measurements of the same individual, using the same tool, under the same conditions (i.e. when we have no reason to expect that the individual being measured would produce a different outcome: [44]). When a measurement tool is used on a single occasion, it is important to know that it will provide an accurate representation of the participant so that the outcome may be used for practical purposes (e.g. diagnostics, or differentiation of individuals or groups). If the same tool is repeatedly used to assess consistency of performance over time, any practice effects need to be understood. This is particularly important as the CFMT is readily available for public access online, and many people who suspect they have DP have often already taken the test prior to formal assessment. If practice effects do occur, alternative versions are needed to assess consistency of performance, allowing for day-to-day fluctuations in cognitive functioning and/or the influence of chance.

Some alternative versions of the CFMT have been created using computer-generated faces, and are typically used for online assessment of self-referring individuals prior to formal in-laboratory testing [45–47]. However, evidence suggests that computer-generated faces are not processed in the same way as natural, real-world faces (e.g. [48–50]) and many authors are reluctant to rely on such tests for formal diagnostics. McKone and colleagues [34] created a new version of the CFMT (the CFMT-Aus) using real-world male faces, and used performance on this second version of the task to clarify 'borderline' cases of DP. The authors also further developed the paradigm by adding an additional 20 s exposure to the target faces at the end of the original test, followed by repeat-administration of the last 54 trials after a 20 min and 24 h delay. While this development clarified the diagnosis of one

of the six individuals that they assessed, it also significantly lengthened the administration time and complexity of the task. These concerns are relevant to repeat administration, particularly when, for practical reasons, multiple attempts at a task may need to be recorded remotely.

Pertinently, there is some evidence to suggest that the original CFMT could be *shortened* by removing the final 24-item 'noise' section. Duchaine & Nakayama [14] originally added noise to the images to force reliance on holistic processing (processing the face as a whole, rather than as a collection of individual features; [51]), although it remains unclear whether (i) the manipulation had the intended consequence, and (ii) whether holistic processing is actually impaired in some/all cases of DP. Further, Corrow *et al.* [52] found that restricting the task to the first 48 trials (i.e. removing the noise section) resulted in very similar diagnostic validity to the entire test, offering a more efficient but equally effective way of diagnosing prosopagnosia, particularly if repeat administration is required.

In the current investigation, we aimed to address the issues reviewed above regarding repeated administration of the CFMT. First, we determined the test–retest reliability of the original version of the task, and investigated whether repeat administration affected performance on the same version. We also subsequently administered an alternative face (the CFMT-Aus) and object (the Cambridge Car Memory Test (CCMT: [43]) version of the task, in order to examine the utility of repeat administration of the same paradigm using novel stimuli. In addition, we examined whether the length of time between testing sessions and testing location (i.e. laboratory versus online) influenced any effects. Finally, we examined whether the trials from the first two sections of the task are equally sensitive to poor performance as the entire test.

# 2. Method

## 2.1. Participants

All participants in this study self-referred to our research team between September 2011 and April 2019, reporting difficulties with face recognition. All reported normal (or corrected to normal) vision, no learning disability or any other neurodevelopmental disorder (autism spectrum condition was excluded using the Autism Quotient: [53]), and no known history of neurological damage or psychiatric illness. Thus, their difficulties were regarded as developmental in origin. All participants are white Caucasian.

Participants took part in the study at two different time points. Based on their geographical location, 30 UK participants (21 female, *M* age at initial testing = 47.9 years, s.d. = 14.32; *M* age at second testing = 52.43, s.d. = 15.05) visited the university for initial testing. These 30 participants were recruited from a pool of 128 individuals (see [21]), who had been screened for DP in person and were invited to take part in the current study. Forty others (32 female, *M* age at initial testing = 52.03 years, s.d. = 14.39; *M* age at second testing = 52.40, s.d. = 14.32) were only assessed online, and were recruited from a pool of 1337 individuals who had registered on our website. Thirty-two participants were from the UK, five the USA, and one from each of Canada, Germany and Austria. No participant withdrew from the study. Details of all participants and their scores can be found in the electronic supplementary material. The two groups did not differ according to age at their initial testing point ($t_{68} = -1.19$, $p = 0.239$) nor at the second ($t_{68} = 0.01$, $p = 0.993$). Informed consent was obtained from all individuals, and all participated on a voluntary basis. The study was carried out in accordance with Bournemouth University Research Ethics Guidelines and was approved by the Bournemouth University Research Ethics Committee.

## 2.2. Materials

*CFMT* [14]: The CFMT uses Caucasian faces taken from the Harvard Face Database (Tong & Nakayama). Images are primarily of Harvard University students: the faces are those of men in their 20s and early 30s, and each individual is displayed in the same range of poses and lighting conditions, with neutral expressions. All faces are cropped so that no hair is visible and facial blemishes are removed. Images are greyscale. The general objective of the CFMT is to introduce six unfamiliar faces to the participant and then test their recognition of those faces. It contains three test stages which increase in difficulty as the test progresses: (i) Learn stage: Participants view a target face from three viewpoints for 3 s per image. They are then presented with triads of faces, where one is an identical image of the target face, and two are distractors. Participants choose which of the three faces is the target. This is repeated for

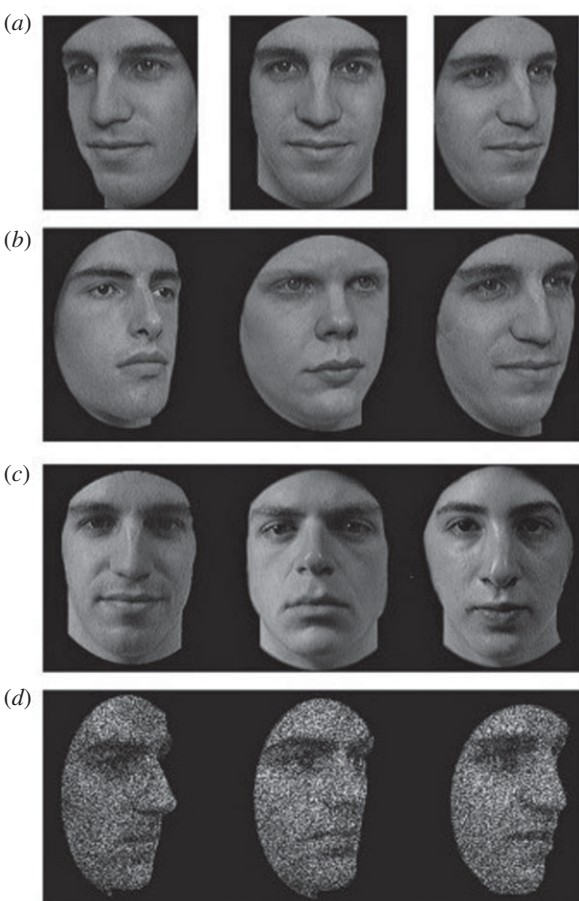

**Figure 1.** Examples of stimuli similar to the CFMT stimuli (note that none of these are items from the test). Panel (*a*) shows the study views: faces are shown one at a time, presented for three seconds each. Panel (*b*) displays a test item from the Learn stage. Here, the face on the right is the same as the studied identity in Panel (*a*). Panel (*c*) shows an item from the Test section (the face on the left is the target). Panel (*d*) displays a test item from the Noise section (face on the right is the target). Image is reproduced from [14] with permission.

six faces, resulting in a maximum score of 18. (ii) Test stage: Thirty triads of faces are presented, where one face is a novel image of a target identity intermixed with two distractors. Target faces are displayed under previously unseen viewpoint or lighting conditions. (iii) Noise stage: Twenty-four new triads are displayed with added visual noise. Again, each trial contains any one of the targets and two distractors. The entire test is scored out of 72, and chance is 24. Example trials of the CFMT can be seen in figure 1.

*CFMT-Aus* [34]: The CFMT-Aus follows the identical format as the original CFMT. Stimuli are also similar in format: faces are greyscale and belong to males aged in their 20s and 30s, with no facial hair or blemishes. The CFMT-Aus is closely matched to the original CFMT with respect to mean score ($M = 80.3\%$, s.d. = 7.6 items and $M = 80.4\%$, s.d. = 7.9, respectively: [14,34]) and internal reliability as measured by Cronbach's alpha ($\alpha = 0.88$ and $\alpha = 0.84$ respectively: [34]). While McKone *et al.* [34] also investigated the utility of repeating the test and noise sections of the task after various delays, this manipulation was not applied here.

*CCMT* [43]: The CCMT is matched in procedure to the CFMT (and, thus, the CFMT-Aus), but simply uses cars instead of faces. Again, this means that better performance is reflected by a higher score, with a maximum score of 72. The stimuli are modified computer-generated images of real-world cars. Similar to the facial stimuli used in the CFMT and CFMT-Aus, cars are presented in greyscale and have no identifying badges, logos, or insignias. During the learning phase in the CCMT, cars are presented side-on, and approximately 30° to the left and 30° to the right. Novel viewpoints and lighting conditions are then applied for the test phase, and noise is added to the stimuli for the final stage, just as in the face tests.

*CFPT* [15]: The CFPT is a computerized sorting task whereby participants are required to organize six faces according to their similarity to a target face. In sum, there are a total of 16 trials: eight upright and

eight inverted. Scores on the CFPT are computed by totalling the deviations from the correct position of each face and so the higher the score the poorer the performance. Totals for the eight upright and eight inverted trials are calculated separately. We only report performance on the upright trials in this paper, following existing literature (e.g. [54,55]), and scores are transferred to proportion correct, rather than raw score errors [16].

*Famous faces:* The famous faces test has previously been used in our published work (e.g. [12,55–57]). Participants were presented with 60 famous faces suitable for the UK population, one at a time, on a computer screen. Each face was cropped so that little extrafacial information was visible and presented in greyscale. Participants were asked to identify the person by either naming them or by providing some autobiographical information unique to that person. At the end, participants were provided with the name of any face they failed to identify; if the participant felt that they were not familiar with that famous person, that trial was eliminated from their final score. Scores are therefore reported as percentages.

## 2.3. Procedure

In an initial testing session, laboratory participants provided demographic information and completed the original version of the CFMT, the CFPT and the famous faces test. The CFMT and CFPT were completed on the same laptop computer under laboratory conditions. To keep administration of the additional tasks consistent with the online group (i.e. so that only the format of the original CFMT was manipulated between the laboratory and online), they were later sent the links to the original CFMT, the CFMT-Aus and the CCMT to complete online in their own time. The tests were completed in a counterbalanced order.

Online participants completed all tasks (including the repeat administration of the original CFMT), with the exception of the famous faces test, via our laboratory's bespoke testing platform (www.prosopagnosiaresearch.org), and provided demographic information by email to the experimenter. This testing platform ensures consistency in the size that images are presented (see [46]), and participants were asked to complete the tests in a quiet environment away from distractions, and not to ask for assistance from another person. All participants completed the tests on a laptop or desktop computer, and tests were completed in a counterbalanced order.

# 3. Results

## 3.1. Data overview

A total of 30 laboratory participants and 40 online participants completed the CFMT on two occasions (from here on referred to as 'CFMT1' for their first attempt at the test, and 'CFMT2' for their second). Of these, 17 laboratory participants (5 male) and 24 online participants (20 female) also completed the CFMT-Aus and CCMT. Three laboratory (one male) and two online participants (both female) completed the CFMT1, CFMT2 and CFMT-Aus. Four laboratory (two male) and five online participants (all female) completed the CFMT1, CFMT2 and CCMT. Six laboratory participants (five female) and nine online participants (five female) completed only the CFMT1 and CFMT2. This information can be seen in figure 2. All laboratory-based participants also completed the famous faces test and CFPT during their initial screening session. Nineteen online participants completed the CFPT.

As many of our laboratory group had completed the test prior to availability of the CFMT online, there was a significant difference between laboratory ($M = 1323.4$ days, s.d. $= 827.2$) and online ($M = 143.5$ days, s.d. $= 78.8$) participants in the time that had elapsed between the first and second testing point: $t_{68} = 8.989$, $p = 0.001$, $d = 2.01$. To explore whether this difference in time influenced any test–retest effects, this factor was entered as a covariate in subsequent group-based analyses (see below).

Initially, we examined the correlations between overall performance on all four tests, collapsed across groups (i.e. all participants: see table 1). Correlations between the CFMT and CCMT were low and non-significant, similar to previous reports [58,59] (see [43,60] for moderate correlations and discussion). The correlation between the CFMT-Aus and CCMT was also low and non-significant as per existing findings [34]. The correlation between repeated administration of the original version of the CFMT indicates its test–retest reliability. Spearman's $r$ was 0.68 ($N = 70$, $p < 0.001$); notably, reliability coefficients of 0.70 are often used as the minimum values to assume test–retest reliability [61,62]. Moreover, participants' performance on the CFMT increased on average by 4.73 correct answers (s.d. $= 8.06$).

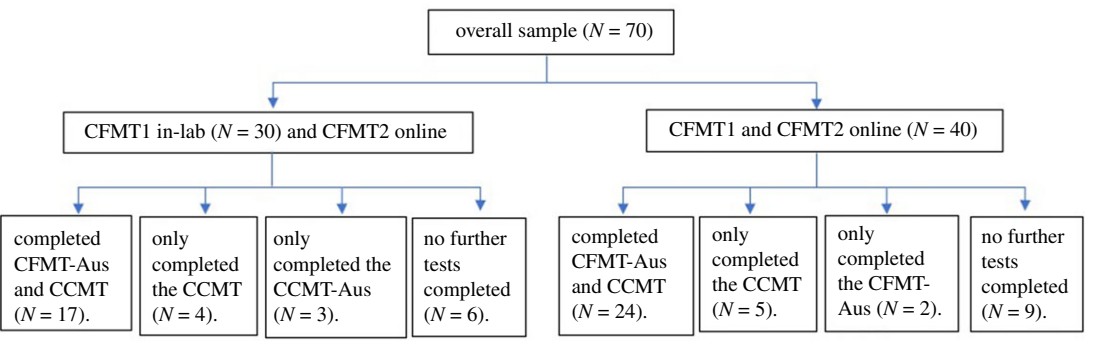

**Figure 2.** Flowchart showing the number of participants who completing each combination of tests.

**Table 1.** Correlation matrix for repeat administration of the original CFMT (total $N = 70$), the CFMT-Aus (total $N = 46$) and the CCMT (total $N = 50$) ($r$ (significant $ps$): the Bonferroni correction for multiple comparisons has been applied).

| test | CFMT2 | CFMT-Aus | CCMT |
| --- | --- | --- | --- |
| CFMT1 | 0.68 (0.001) | 0.58 (0.001) | −0.20 |
| CFMT2 | | 0.59 (0.001) | 0.02 |
| CFMT-Aus | | | 0.04 |

To further assess the diagnostic utility of different sections of the CFMT, we correlated performance on each section of the original test against alternative DP indicators: the CFPT and famous faces test (table 2). Only laboratory-based participants completed the famous faces test. All laboratory participants and 19 online participants completed the CFPT (see electronic supplementary material for more details and scores). CFPT scores did not correlate with any stage of the CFMT, supporting the distinction between face perception and face memory and, consequently, existing models of face processing ([63]; but see [2,37,64]). Interestingly, famous face performance correlated significantly with the learn and test stages of the CFMT, but not the noise section.

## 3.2. Group analyses

A 4 (test: CFMT1, CFMT2, CFMT-Aus, CCMT) x 3 (section: learn, test, noise) x 2 (group: lab, online) ANCOVA (controlling for time elapsed between first and second testing session) examined performance on specific sections of the tests. The three-way interaction was non-significant: $F_{6,228} = 1.008$, $p = 0.421$. There was a significant interaction between test and section, superseding a main effect of section: $F_{6,228} = 7.567$, $p = 0.001$, $\eta\rho^2 = 0.166$ and $F_{2,76} = 177.747$, $p = 0.001$, $\eta\rho^2 = 0.824$, respectively. Follow-up contrasts (Bonferroni corrected) indicated that for the learn phase, performance in the CFMT1 was lower than the CFMT2 ($p = 0.017$), and performance in both the CFMT2 and CFMT-Aus was higher than the CCMT ($ps < 0.001$). For the test phase, performance in the CFMT1 was lower than both the CFMT2 and CFMT-Aus ($ps < 0.001$), and performance in the CCMT was lower than the CFMT2 ($p = 0.010$) and CFMT-Aus ($p = 0.002$). For the noise section, performance in all three face tests was lower than in the cars test (all $ps < 0.001$), and performance on the CFMT-Aus was lower than the CFMT2 ($p = 0.008$). These findings indicate a practice effect in the learn and test phases of the original CFMT, but not the noise section. For a summary of the descriptive statistics, see table 3.

The interactions between test and group and between section and group, and the main effects of test and group, were all non-significant: $F_{3,114} = 1.830$, $p = 0.146$; $F_{2,78} = 0.677$, $p = 0.511$; $F_{3,117} = 2.473$, $p = 0.065$; and $F_{1,39} = 0.103$, $p = 0.750$, respectively. There was no influence of the covariate (all $ps > 0.05$).

## 3.3. Single case analyses

To assess whether practice effects on the CFMT can influence diagnostic outcomes, we compared individual participants' CFMT1 and CFMT2 scores at the single case level. We use the norms from the original CFMT publication [14] to identify atypical scores: scores of 42 and below ($M = 57.9$, s.d. $= 7.9$).

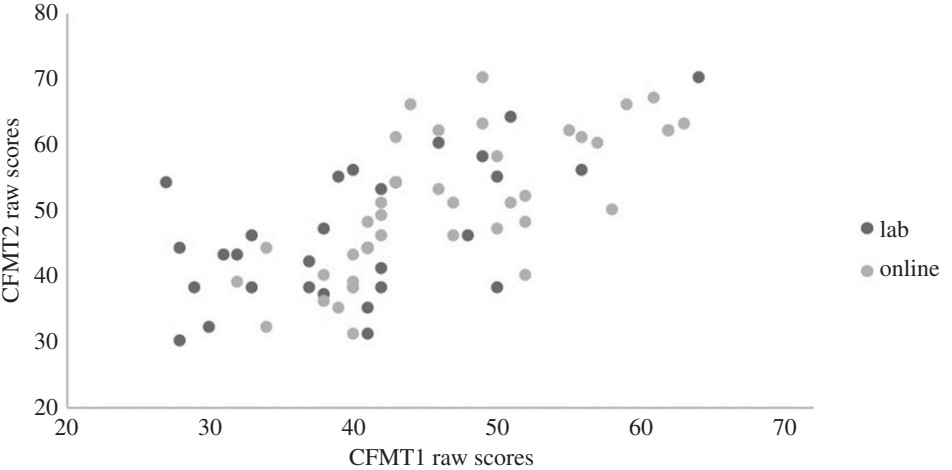

**Figure 3.** Raw scores (out of a possible total of 72) on the CFMT1 and CFMT2 for each participant, indicating the different testing modes for CFMT1 (i.e. in laboratory or online; all CFMT2 completions were online).

**Table 2.** Correlation matrix for each section of the original CFMT1 and the CFPT and famous faces tasks. Note that not all laboratory participants completed all three tests. Nineteen of the online participants completed the CFPT (total $N = 49$); none of the online participants completed the famous faces test (total $N = 30$) ($r$ (significant $ps$): the Bonferroni correction for multiple comparisons has been applied).

| test: stage | CFMT: test | CFMT: noise | CFPT | famous faces |
|---|---|---|---|---|
| CFMT: learn | 0.44 (0.001) | 0.41 (0.001) | 0.16 | 0.46 (0.011) |
| CFMT: test | | 0.54 (0.001) | 0.15 | 0.53 (0.003) |
| CFMT: noise | | | 0.17 | 0.30 |

**Table 3.** Descriptive statistics for the separate stages of the CFMT1, CFMT2 and CFMT-Aus, presented as raw scores.

| test | test stage: $M$ (s.d.) | | |
|---|---|---|---|
| | learn | test | noise |
| CFMT1 | 16.25 (2.18) | 16.61 (5.05) | 11.16 (4.00) |
| CFMT2 | 16.73 (1.45) | 19.07 (5.62) | 11.94 (4.36) |
| CFMT-Aus | 16.48 (1.82) | 19.07 (5.00) | 9.72 (3.32) |
| CCMT | 15.20 (1.71) | 15.76 (3.85) | 14.96 (3.29) |

These norming data and the recommended cut-off score have been echoed in larger, heterogenous samples (e.g. [2]; $M = 58.2$, s.d. $= 8.3$). For CFMT1, 21 laboratory participants were impaired. However, 10 of the 21 laboratory participants would no longer meet the diagnostic criteria for DP when assessing CFMT2 scores (i.e. they scored within the typical range of scores when tested a second time; $M$ improvement $= 13.40$, s.d. $= 6.20$). One individual scored within the DP range on CFMT2 but not on CFMT1 (31/72 and 50/72 respectively). For the online participants, sixteen met the criteria for DP on CFMT1, but seven of these no longer met this criterion on CFMT2 ($M$ improvement $= 6.14$, s.d. $= 2.85$). Similarly to the laboratory group, there was one participant who scored within the DP range on CFMT2 but not on CFMT1 (40/72 and 52/72 respectively). Further, an additional 18 participants (six laboratory participants, 12 online participants) scored better at CFMT1 than CFMT2, indicating some inconsistency in performance aside from practice effects. These data can be seen in figure 3.

To further explore this, we also compared CFMT1 and CFMT-Aus scores at the individual level. Nineteen laboratory participants completed the CFMT-Aus. Fourteen scored atypically on CFMT1 and six of these individuals also scored atypically on the CFMT-Aus. Twenty-six of the online participants

completed the CFMT-Aus. Of these, 11 scored atypically on CFMT1, and seven of these also scored atypically on the CFMT-Aus. The individual who scored within the DP range on CFMT2 but not CFMT1, also scored atypically on the CFMT-Aus, supporting the necessity for repeated assessment in some individuals. Three participants scored typically on both CFMT1 and CFMT2, but scored atypically on the CFMT-Aus.

Finally, to consider the issue of domain-specificity, we also considered the CCMT as a measure of object recognition ability in terms of the number of intact or impaired cases. Thus, we examined the participants' performance on these three tests at the individual level. To disentangle any test–retest argument and/or practice effects with specific stimuli, we only address the CFMT1, CFMT-Aus and CCMT scores (i.e. we do not look at the participants' second attempt on the CFMT, i.e. CFMT2). We use the norms from the original publications to identify atypical scores. For the CFMT, these are scores of 42 and below ($M =$ 57.9, s.d. = 7.9), and for the CFMT-Aus, these are scores of 43 and below ($M = 57.7$, s.d. = 7.3). Because the CCMT elicits strong gender effects, we used separate cut-offs for atypical performance in females (less than 37; $M = 50.4$, s.d. = 7.2) and males (less than 41; $M = 57.4$, s.d. = 8.3).

Twenty-one laboratory participants completed the CFMT1 and the CCMT. CFMT and CCMT scores did not significantly correlate with each other in this sample ($r = -0.43$, $p = 0.051$). Of these, 15 scored atypically on the CFMT1, but only two scored within the impaired range on the CCMT (one female scored 33, one male scored 39). Neither of these participants scored atypically on the CFMT-Aus (scores of 45 and 48 respectively).

Twenty-nine online participants completed the CFMT1 and CCMT. CFMT and CCMT scores did not significantly correlate with each other in this sample ($r = 0.27$, $p = 0.162$). Of these, 12 scored atypically on the CFMT1, including two who scored in the impaired range on the CCMT (both female, scoring 36 and 32). One of these participants did not complete the CFMT-Aus. The other participant, however, also scored atypically on the CFMT-Aus (i.e. scored atypically on all three tests). Of the 23 participants who scored atypically on CFMT1 but scored within the typical range on the CCMT, standardized difference tests [65] revealed significant dissociations between face and object memory in nine individuals. These data are summarized in table 4. Note that for these calculations, control data were taken from the associated publications [14,43] and correlation coefficients were taken from Dennett *et al.* [43].

Finally, two additional online participants scored atypically on the CCMT (both female, scoring 35 and 36), but scored typically on the CFMT1 (scoring 58 and 51 respectively) and CFMT-Aus (scoring 53 and 48 respectively). Cases where scores on the CCMT are atypical are summarized in table 5, and a full set of scores can be found in the electronic supplementary material.

## 3.4. A shortened CFMT

It was also of interest to explore the diagnostic utility of the noise section of the CFMT. In the above analyses we found that this section is the only section that is not subject to a practice-effect, yet it is also the only section that does not correlate with famous face recognition performance. It has recently been revealed that the noise section of the CFMT may not enhance the diagnostic utility of the test overall and that using only the first 48 trials results in an equally effective instrument for diagnosing DP [52]. Thus, we compared whether the individuals who met the diagnostic criteria for DP on the full CFMT (first attempt; score of 42 and below) also met the diagnostic criteria for DP using only the 'learn' and 'test' subsections (i.e. the first 48 trials). We use the cut-off for atypical performance provided by Corrow *et al.* [52] of 33 or fewer correct responses.

Thirty-seven of the 70 participants scored 42 or less on the CFMT1. Of these, 36 would also have scored atypically if we had only considered their scores on the first 48 trials. Interestingly, three participants scored atypically on the short version on the CFMT, but did not when scores on all three sections were assessed. Thus, it may be that in some cases, those with self-reported face recognition difficulties perform particularly well on the noise section on the CFMT, resulting in a typical score.

# 4. General discussion

This study examined repeated administration of the CFMT in a group of individuals who self-referred to our research team complaining of face recognition deficits ($N = 70$). For the first known time, we report the test–retest reliability of the test, considering testing location (online or in the laboratory) and the time lapse between attempts. The diagnostic utility of administering an alternative version of the CFMT was

**Table 4.** The results of the revised standardized difference test (RSDT) that investigated whether standardized differences between the participants' scores on the CFMT1 and CCMT are significantly different from controls. Reported statistics represent *t*-values; the estimated percentage of the control population exhibiting a difference more extreme than the individual is presented in parentheses. Twenty-three participants scored atypically at CFMT1 and typically on CCMT. The standardized difference between these scores for nine participants significantly differed from controls.

| participant code | CFMT1 raw score | CCMT raw score | RSDT result |
| --- | --- | --- | --- |
| Lab_F02 | 33 | 62 | 4.36 (0.00)** |
| Lab_F04 | 42 | 43 | 0.90 (18.46) |
| Lab_F06 | 33 | 56 | 3.60 (0.03)* |
| Lab_F07 | 38 | 45 | 1.62 (5.42) |
| Lab_F08 | 37 | 41 | 1.23 (11.13) |
| Lab_F09 | 41 | 59 | 3.05 (0.15)* |
| Lab_F12 | 32 | 61 | 4.35 (0.00)* |
| Lab_F13 | 40 | 41 | 0.88 (19.06) |
| Lab_M01 | 29 | 59 | 3.45 (0.06)* |
| Lab_M02 | 31 | 59 | 3.23 (0.11)* |
| Lab_M04 | 28 | 46 | 2.16 (1.77)* |
| Lab_M06 | 37 | 55 | 2.12 (1.98)* |
| Lab_M09 | 41 | 47 | 0.80 (21.51) |
| Online_F02 | 40 | 43 | 1.13 (12.98) |
| Online_F03 | 32 | 44 | 2.19 (1.55)* |
| Online_F05 | 40 | 47 | 1.64 (5.18) |
| Online_F06 | 34 | 40 | 1.45 (7.54) |
| Online_F08 | 42 | 46 | 1.28 (10.11) |
| Online_F23 | 41 | 42 | 0.89 (18.76) |
| Online_F27 | 41 | 44 | 1.15 (12.74) |
| Online_M01 | 38 | 41 | 0.49 (31.40) |
| Online_M02 | 38 | 44 | 0.81 (21.04) |
| Online_M04 | 42 | 41 | 0.03 (48.69) |

*$p < 0.05$, **$p < 0.001$.

**Table 5.** Individual scores for the six participants who scored atypically on the CCMT: these are scores of 36 and below for females and 40 and below for males. Asterisks denote cases where there is a reversed pattern of impairment. That is, these individuals have object recognition deficits but typical face recognition abilities (as assessed by these three tasks).

| participant code | CFMT1 | CFMT-Aus | CCMT |
| --- | --- | --- | --- |
| Lab_F03 | 39 | 45 | 33 |
| Lab_M03 | 42 | 48 | 39 |
| Online_F01 | 34 | not completed | 36 |
| Online_F20 | 40 | 38 | 32 |
| Online_F09* | 58 | 53 | 35 |
| Online_F10* | 51 | 48 | 36 |

also explored (the CFMT-Aus), and the issue of domain-specificity was examined via an object version of the same paradigm (the CCMT). Finally, the efficacy of a shortened version of the CFMT was evaluated.

Importantly, overall analyses indicated there was no correlation between CCMT scores and any CFMT test (i.e. CFMT1, CFMT2 and the CFMT-Aus), supporting previous demonstrations of content

validity [43]. However, the test–retest reliability of the CFMT was found to fall just short of accepted psychometric standards, producing a reliability coefficient of 0.68. Practice effects were observed for the 'learn' and 'test' stage of the task, and there was no influence of the time that had elapsed between attempts. Single-case analyses supported this finding, where 37 of the 70 participants scored below cut-off on their first attempt at the CFMT, but only 17 scored atypically on their second attempt. This finding of a practice effect is important not only for diagnostic purposes, but also for training programmes that rely on repeat-administration of the CFMT to monitor the efficacy of the training regime [66–68].

Pertinently, two participants scored within the typical range of scores on the CFMT1 but atypically on their second attempt. Eighteen other participants (six laboratory, 12 online) scored better at CFMT1 than CFMT2, indicating some day-to-day inconsistencies in performance aside from practice effects. These inconsistencies, and the practice effect itself, may result from individual variability in fluctuations in cognitive performance [26–30], or from intrinsic factors such as mood, fatigue, or distraction. While we did not directly assess these factors here, they should be considered in future research. Alternatively, this pattern of findings may simply reflect the influence of chance [20,34]. Unless a task has perfect reliability, individual scores suffer from at least some uncertainty. Yet, this measurement error is rarely taken into account by researchers when identifying cases of DP. Indeed, some individuals may meet the diagnostic criteria for DP as measured by a single attempt on a test on one day, yet be identified as 'typical' on another attempt, or on another test (or vice versa), purely due to statistical chance (for a full discussion and example cases, see [34]). To some extent this is offset by diagnostic approaches that require impaired performance on multiple tests within the same session (e.g. [69–71]), although the most popular tasks (CFMT, CFPT and famous faces) arguably tap different sub-processes of face recognition that can be selectively impaired in some individuals. Thus, the issue of repeat assessment on different versions of the same test is particularly pertinent in such cases.

Additionally, these inconsistencies may be due to the testing location; that is, data from online samples may involve a trade-off between participant numbers and data quality (see [47] for a discussion). However, we report no significant difference in test–retest effects between those who completed the CFMT for the first time in the laboratory, and those who completed all the tasks online. In support of this, existing reports show that there are no significant differences in performance on face-processing tasks completed online and in the laboratory (e.g. [32,47]). Thus, online screening for DP is probably just as reliable as in-laboratory testing, and is of course more efficient given the large numbers of people that self-report for assessment. However, one caveat of online testing is that some tasks are openly available online, including the CFMT. Thus, when participants self-refer to more than one laboratory, it is possible that they have previously completed dominant tasks on at least one occasion. Consequently, researchers should enquire whether their participants have previously completed the CFMT, and also provide individual participants with their scores so these can be presented to other laboratories on request.

The importance of data sharing is also likely to be valid for the typical population. While this investigation aimed to evaluate the influence of test–retest effects on the diagnosis of prosopagnosia, it is likely that a practice effect also presents for the CFMT in typical perceivers. Indeed, both the CFMT and its extended version (the CFMT+: [64]) are frequently used to assess individual differences in typical face recognition ability, and future work should investigate test–retest reliability in this population.

A potential solution to practice effects on the CFMT is to develop different versions of the task, using the same paradigm but novel stimuli. The current study investigated this possibility, via administration of the CFMT-Aus (an alternative test of face memory, which is matched in format and procedure to the original CFMT) after the first attempt at the original CFMT. At the individual level, we reported that 13 participants (out of 46 participants who completed both tasks) scored atypically on the CFMT1 and the CFMT-Aus. One individual scored within the DP range on the CFMT2 and CFMT-Aus, but not on their first attempt on the CFMT. Moreover, six participants scored atypically on their first attempt at the CFMT but scored within the typical range on their second attempt. An atypical score on the CFMT-Aus confirmed their difficulties. Taken together, these findings support the case for repeated assessment, to establish a reliable baseline of ability. Further, as there were inconsistencies in performance between the CFMT1 and CFMT-Aus in both directions, we recommend that repeat assessment should occur on a separate day. This would address any ambiguous cases stemming from differences in day-to-day cognitive functioning.

Notably, our findings also indicate a potential practice effect for the test phase of the CFMT-Aus. That is, performance on CFMT-Aus test trials was higher than the performance on CFMT1 test trials (but this pattern was not observed for learn nor noise trials). While further work is required to confirm whether this finding is genuinely a practice effect (i.e. by manipulating order of presentation of the two tests), it is

possible that previous experience with the CFMT paradigm provides prior knowledge that can aid the use of compensatory strategies in future attempts at the task, even when novel stimuli are used. Interestingly, the same effect was not observed for the CCMT, perhaps because compensatory strategies for face recognition do not extend to the recognition of cars.

However, it should be noted that performance on the noise section of the CCMT was higher than in the three face versions. The lack of a practice effect in this section for the face versions of the task may indicate that the addition of noise differentially affects face versus object processing. That is, the reason that no practice effect emerged in the noise section for faces is that the addition of noise removes featural information and forces a reliance on holistic processing [72]. Because our participants all self-reported with face recognition difficulties, it follows that the addition of noise prohibits the use of feature-by-feature compensatory strategies, and exacerbates difficulties in those individuals that struggle with the holistic processing of faces [73–75]. In contrast, because holistic processing is not thought to benefit the recognition of objects such as cars [76–78], the addition of noise to these stimuli is not so detrimental in our population.

This hypothesis is supported by our analyses that directly addressed the domain-specificity of the face recognition difficulties reported by our population—a question that has received much attention in very recent years [16,79,80]. In the current study, two of the 15 laboratory participants who completed the CFMT1 and the CCMT were impaired on both tests, as were two of the 12 online participants who completed these tasks. At first glance, these data suggest that a much smaller proportion (15%) of individuals with face recognition difficulties have comorbid object recognition deficits than reported in previous work. While this substantially differs to the projected figure of 80% of all DP cases offered by Geskin & Behrmann [79], we acknowledge that we only tested one category of object, as opposed to the recommendation of assessing multiple object categories [81,82]. Further, individual expertise with cars can greatly vary, and their within- and between-exemplar visual properties probably differ from faces [83].

Finally, the CFMT's noise trials have also recently been considered in light of findings that the test retains its sensitivity when only the learn and test stages are administered [52]. This suggestion is again supported by the finding reported here that the recognition of famous faces only correlates with the first two stages of the CFMT, and not the noise section. We therefore further assessed the diagnostic utility of the CFMT when the noise section was removed; of the 37 participants who scored atypically on the CFMT overall, 36 of these would also be considered for a DP diagnosis if performance on only the first two stages were measured. While this finding supports a more efficient, shorter version of the task that will benefit the burden of repeated assessment, it should be noted that three participants scored atypically on the short version of the CFMT, but did not when scores on all three sections were assessed. It is unclear why this finding emerged, although previous work indicates that holistic processing is not always impaired in people with face recognition difficulties (e.g. [84–86]).

## 5. Conclusion

At present, individuals are screened for DP by administering objective face-processing tests (typically the CFMT) on a single occasion. However, as performance on these tests can be affected by factors irrespective of face-processing skills (e.g. mood, fatigue, chance) and the test–retest reliability of the CFMT falls just short of the accepted parametric standards, we recommend a new protocol for the screening of DP: that a second, alternative test which assesses the same skill—in this case, unfamiliar face memory—should be administered, preferably on a different day. To make this process more efficient, our data support the claim that the CFMT can be shortened; the learn and novel stages are as sensitive to identifying cases of DP as is the entire test.

Ethics. The study was carried out in accordance with Bournemouth University Research Ethics Guidelines and was approved by the Bournemouth University Research Ethics Committee (reference ID: 21080).

Data accessibility. The datasets generated and analysed during the current study are uploaded as electronic supplementary material (Excel Spreadsheet).

Authors' contributions. Conceived and designed the experiments: E.M. and S.B. Data collection: E.M. Analysed the data: E.M. Contributed reagents/materials/analysis tools: E.M. and S.B. Draft preparation and editing: E.M. and S.B. Both authors gave final approval for publication and agree to be held accountable for the work performed therein.

Competing interests. We declare we have no competing interests.

Funding. S.B. is supported by a British Academy Mid-Career Fellowship (grant no. MD170004).

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
