## [Reviewer comments · Royal Society Open Science]

Review History

RSOS-200884.R0 (Original submission)

Review form: Reviewer 1

Is the manuscript scientifically sound in its present form?

Yes

Are the interpretations and conclusions justified by the results?

Yes

Is the language acceptable?

Yes

Do you have any ethical concerns with this paper?

No

Have you any concerns about statistical analyses in this paper?

No

Recommendation?

Accept with minor revision (please list in comments)

Comments to the Author(s)

While the CFMT has been used extensively in Psychology to screen for Developmental Prosopagnosia, its test-retest reliability has not been examined. This is an important omission that is addressed by this paper. Performance on the CFMT is compared at different time points and also for different contexts (laboratory and online) and contrasted with several other appropriate – and commonly used – comparisons (CFPT, famous face recognition test, CCMT). The results show that test-retest reliability of the CFMT falls just short of psychometric standards at .68. Important individual cases are noted – for example, of the laboratory participants that were impaired on the CFMT at time 1, almost half (10/21) did not meet these criteria at time 2. The paper also presents comparison with the CFMT-Aus, investigates practice effects, and looks at shortening of the CFMT.

I read this work with great interest and, in my view, it is publishable in its present form. Test-retest reliability has been ignored for far too long in this domain, but is becoming more pertinent than ever. The importance of understanding test-retest reliability is laid out clearly in the introduction and motivated well. The methods are sound and the analyses are appropriate and carried out carefully. The results section is extensive and requires some careful focused reading. I would not cut it down but wonder whether the Discussion can be shortened to reduce some repetition here. And on a very minor note, I could not see any cases indexed by an Asterisk in Table 5 in my copy of manuscript.

Review form: Reviewer 2 (Meike Ramon)

Is the manuscript scientifically sound in its present form?

Yes

Are the interpretations and conclusions justified by the results?

No

Is the language acceptable?

Yes

Do you have any ethical concerns with this paper?

No

Have you any concerns about statistical analyses in this paper?

No

Recommendation?

Accept with minor revision (please list in comments)

Comments to the Author(s)

See comments attached (Appendix A).

Decision letter (RSOS-200884.R0)

Dear Ms Murray

On behalf of the Editors, I am pleased to inform you that your Manuscript RSOS-200884 entitled "Diagnosing Developmental Prosopagnosia: Repeat Assessment using the Cambridge Face Memory Test" has been accepted for publication in Royal Society Open Science subject to minor revision in accordance with the referee suggestions. Please find the referees' comments at the end of this email.

The reviewers and handling editors have recommended publication, but also suggest some minor revisions to your manuscript. Therefore, I invite you to respond to the comments and revise your manuscript.

- Ethics statement

- Data accessibility

If you wish to submit your supporting data or code to Dryad (<http://datadryad.org/>), or modify your current submission to dryad, please use the following link:
<http://datadryad.org/submit?journalID=RSOS&manu=RSOS-200884>

- Competing interests

- Authors' contributions

- Acknowledgements

- Funding statement

Because the schedule for publication is very tight, it is a condition of publication that you submit the revised version of your manuscript before 29-Jul-2020. Please note that the revision deadline will expire at 00.00am on this date. If you do not think you will be able to meet this date please let me know immediately.

Supplementary files will be published alongside the paper on the journal website and posted on the online figshare repository (<https://rs.figshare.com/>). The heading and legend provided for each supplementary file during the submission process will be used to create the figshare page,

so please ensure these are accurate and informative so that your files can be found in searches. Files on figshare will be made available approximately one week before the accompanying article so that the supplementary material can be attributed a unique DOI.

If your manuscript is newly submitted and subsequently accepted for publication, you will be asked to pay the article processing charge, unless you request a waiver and this is approved by Royal Society Publishing. You can find out more about the charges at <https://royalsocietypublishing.org/rsos/charges>. Should you have any queries, please contact openscience@royalsociety.org.

on behalf of Dr Bruno Rossion (Associate Editor) and Essi Viding (Subject Editor)
openscience@royalsociety.org

Reviewer comments to Author:
Reviewer: 1

Comments to the Author(s)

While the CFMT has been used extensively in Psychology to screen for Developmental Prosopagnosia, its test-retest reliability has not been examined. This is an important omission that is addressed by this paper. Performance on the CFMT is compared at different time points and also for different contexts (laboratory and online) and contrasted with several other appropriate – and commonly used – comparisons (CFPT, famous face recognition test, CCMT). The results show that test-retest reliability of the CFMT falls just short of psychometric standards at .68. Important individual cases are noted – for example, of the laboratory participants that were impaired on the CFMT at time 1, almost half (10/21) did not meet these criteria at time 2. The paper also presents comparison with the CFMT-Aus, investigates practice effects, and looks at shortening of the CFMT.

I read this work with great interest and, in my view, it is publishable in its present form. Test-retest reliability has been ignored for far too long in this domain, but is becoming more pertinent than ever. The importance of understanding test-retest reliability is laid out clearly in the introduction and motivated well. The methods are sound and the analyses are appropriate and carried out carefully. The results section is extensive and requires some careful focused reading. I would not cut it down but wonder whether the Discussion can be shortened to reduce some repetition here. And on a very minor note, I could not see any cases indexed by an Asterisk in Table 5 in my copy of manuscript.

Reviewer: 2

Comments to the Author(s)

See comments attached

Author's Response to Decision Letter for (RSOS-200884.R0)

See Appendix B.

Decision letter (RSOS-200884.R1)

Dear Ms Murray,

It is a pleasure to accept your manuscript entitled "Diagnosing Developmental Prosopagnosia: Repeat Assessment using the Cambridge Face Memory Test" in its current form for publication in Royal Society Open Science.

on behalf of Dr Bruno Rossion (Associate Editor) and Professor Essi Viding (Subject Editor)
openscience@royalsociety.org

Appendix A

Review of RSOS-200884

The authors report an investigation into aspects concerning administration of the CFMT and other tools for the diagnosis of developmental prosopagnosia (DP). The authors determine the CFMT's (1) test-retest reliability (across time and administration modus), (2) relationship to procedurally identical, yet independent measures of face and object processing, (3) diagnostic value when 1/3 of trials are omitted. We appreciate the motivation for this study, which addresses important issues in DP research, and find the manuscript well written. We recommend publication after addressing the issues detailed below. We hope the authors will find our comments constructive and remain at their disposal for questions.

Sincerely,

Meike Ramon and Christopher Turner

Major

1. Subject selection and identification as DPs

The authors state that “All participants in this study self-referred to the research team reporting difficulties with face recognition.” One would assume that over 8 years, more than 70 people will have contacted the lab, or accessed www.prosopagnosiaresearch.org. Please describe in detail the criteria applied to select individuals of the two subsets reported here, and if any individuals withdrew from the study. Provide a summary of individual scores across all tests, and refer to any previous publication of the cases and use consistent acronyms to allow cross-study comparison. This aligns with the authors emphasis of the importance of data sharing (p.18f.)

Considering the authors aimed to address “the diagnostic utility of administering” different tests, We would like to see the previous formal criteria used for DP diagnosis contrasted with those proposed based on their findings. Specifically, in our opinion, readers will be interested in seeing the differences in DP across both sets of criteria.

Note also that here norms of the original CFMT publication were used to identify atypical scores, which were collected in the lab. Based on previous findings, it is reasonable to assume that this original sample would have produced data suggesting a higher cut-off (cf. e.g. CFMT scores reported by Bobak et al., 2016, *Frontiers in Psychology* vs. Stacchi et al., 2020). Given that there is now ample data from the CFMT(+), please provide comparative data from a larger, more heterogeneous sample as well. This could yield substantially

different classification than provided on p.14, l.7ff. (21/30 of lab-tested and 16/40 of online-tested Ss below the cut-off).

2. *Data presentation / comprehensibility*

The relationships between individual test performance across various measures are very interesting, but we found it hard to integrate based on how the data are currently provided. Please represent the (individual) data in a more reader-friendly, visual way that allows a direct comparison of scores across different modes of testing and tests.

3. *Measures subject to analyses*

P.21, l.12ff.: in addition to considering multiple object categories, note that Geskin & Behrmann's findings resulted from parallel consideration of accuracy *and* RTs, which were not considered here. Please provide analyses related to this additional measure, which has proven to be highly informative when it comes to impaired populations (cf eg Delvenne et al. 2004; Michel & Rossion, 2018).

Minor

- For the less informed reader, please include a short terminological distinction between the different sub-processes involved in face processing in relation to each test used. This is not related to the content of this manuscript, but rather to address the conceptual confusion propagated by others, who you adopt a less stringent approach to definitions and concepts.
- The two groups did not differ according to age at their initial testing point [...] nor at the second – would we expect them to?
- Include a visualization of the different tests, as well as a graphical summary / diagram of the information provided in the “Data overview” section.
- p.12, l.44ff.: “CFPT scores did not correlate with any stage of the CFMT, supporting the distinction between face perception and face memory and, consequently, existing models of face processing [63].” There are multiple tests of face perception that actually do correlate with the CFMT. We would like to see this acknowledged, as well as a discussion of the CFPT's utility.
- Provide demographic information for the subjects and include how it was recorded in the procedure. (Subject location and ethnicity is alluded to in the text but this information is not provided.)
- Clarify the Table 5 note “* denotes cases where there is a reversed pattern of impairment”, as there is no asterisk (“*”) in the table.

Appendix B

We wish to thank both reviewers for their comments and feedback, and we are especially pleased to hear that Reviewer 1 believes the manuscript is publishable in its present form. We have made some adjustments to the manuscript to address the constructive comments provided by the reviewers, and we detail these below.

- **Please describe in detail the criteria applied to select individuals of the two subsets reported here, and if any individuals withdrew from the study. Provide a summary of individual scores across all tests, and refer to any previous publication of the cases and use consistent acronyms to allow cross-study comparison.**

This information has now been included in the Participants subsection, P.7. We have pointed readers to an existing publication, stated that no participants withdrew, and have detailed that all scores and details of the participants can be found in the Supplementary Information.

- **We would like to see the previous formal criteria used for DP diagnosis contrasted with those proposed based on their findings. Specifically, in our opinion, readers will be interested in seeing the differences in DP across both sets of criteria.**

Please note that existing DP diagnostic criteria is based on performance across multiple tests (including the CFPT and famous face recognition tests), and not just the CFMT. This paper only examines performance on the CFMT and how repeat testing is important to decipher whether or not a participant is impaired on this particular task. Because of this, and because not all participants took part in the CFPT and famous faces test that are required for a complete diagnostic profile, we cannot address the issue of wider patterns of performance here. We have made it clearer throughout the manuscript (see *Data Overview*, and P.12) that online participants did not complete the famous faces test, and only 19 of these participants completed the CFPT.

- **Given that there is now ample data from the CFMT(+), please provide comparative data from a larger, more heterogeneous sample as well. This could yield substantially different classification than provided on p.14.**

We thank you for this recommendation and have included information on P. 13 to address this. Data from a larger sample, such as that of Bowles et al. (2009), echoed that of the original publication and also suggested a cut-off of 42 and below. With this in mind, the present data do not change and no further edits are made.

- **Please represent the (individual) data in a more reader-friendly, visual way that allows a direct comparison of scores across different modes of testing and tests**

Thank you for this recommendation. We have now included a Figure which compares CFMT1 and CFMT2 scores, for lab-based participants and online participants.

- **In addition to considering multiple object categories, note that Geskin & Behrmann's findings resulted from parallel consideration of accuracy and RTs, which were not considered here. Please provide analyses related to this additional measure, which has proven to be highly informative when it comes to impaired populations**

Thank you for this comment. While we appreciate the importance of assessing RTs as well as accuracy, particularly for face perception tasks (e.g. Rossion & Michel, 2018), RTs are not assessed in the CFMT memory-based paradigm. Further, because we used the standard instructions of the task, participants were not informed that RTs would be analysed. Therefore, we have no reliable basis to do so here.

- **For the less informed reader, please include a short terminological distinction between the different sub-processes involved in face processing in relation to each test used.**

This information can already be found on P.3-4 in the Introduction.

- **The two groups did not differ according to age at their initial testing point [...] nor at the second –would we expect them to?**

Yes, this is possible as we did not impose a consistent time period between participants' first and second attempts. This information is included on P.11 where it is also explained that the time-lapse is a necessary covariate in the analyses.

- **Include a visualization of the different tests, as well as a graphical summary / diagram of the information provided in the “Data overview”**

We have now included a visual for an example trial of the original CFMT (Figure 1). There is also a visualisation of the data overview section as a flowchart, now provided as Figure 2.

- **There are multiple tests of face perception that actually do correlate with the CFMT. We would like to see this acknowledged, as well as a discussion of the CFPT’s utility.**

We have now directed readers to a selection of papers which did find a correlation between tests of face perception and the CFMT on P.11. However, we have not included a discussion of the CFPT’s utility as we believe this is out of the scope of the present paper. Furthermore, Reviewer 1 has suggested that the Discussion section is reduced in length and we therefore prioritise other information there.

- **Provide demographic information for the subjects and include how it was recorded in the procedure. (Subject location and ethnicity is alluded to in the text but this information is not provided.)**

Thank you for bringing this to our attention. We have now included demographic information in the *Participants* subsection, and stated how it was collected in the *Procedure*.

- **Clarify the Table 5 note “* denotes cases where there is a reversed pattern of impairment”, as there is no asterisk (“*”) in the table.**

We thank the reviewers for identifying this and we have now amended the Table.